# Targeting Asparagine Synthetase in Tumorgenicity Using Patient-Derived Tumor-Initiating Cells

**DOI:** 10.3390/cells11203273

**Published:** 2022-10-18

**Authors:** Gen Nishikawa, Kenji Kawada, Keita Hanada, Hisatsugu Maekawa, Yoshiro Itatani, Hiroyuki Miyoshi, Makoto Mark Taketo, Kazutaka Obama

**Affiliations:** 1Department of Gastrointestinal Surgery, Graduate School of Medicine, Kyoto University, Kyoto 606-8507, Japan; 2Department of Surgery, Kyoto City Hospital, Kyoto 604-8845, Japan; 3Department of Surgery, Rakuwakai Otowa Hospital, Kyoto 607-8062, Japan; 4Institute for Advancement of Clinical and Translational Science (IACT), Kyoto University Hospital, Kyoto 606-8507, Japan

**Keywords:** asparagine synthetase, asparagine metabolism, L-asparaginase, KRAS-driven cancer, patient-derived spheroid, patient-derived spheroid xenograft

## Abstract

Reprogramming of energy metabolism is regarded as one of the hallmarks of cancer; in particular, oncogenic RAS has been shown to be a critical regulator of cancer metabolism. Recently, asparagine metabolism has been heavily investigated as a novel target for cancer treatment. For example, Knott et al. showed that asparagine bioavailability governs metastasis in a breast cancer model. Gwinn et al. reported the therapeutic vulnerability of asparagine biosynthesis in KRAS-driven non-small cell lung cancer. We previously reported that *KRAS*-mutated CRC cells can adapt to glutamine depletion through upregulation of asparagine synthetase (ASNS), an enzyme that synthesizes asparagine from aspartate. In our previous study, we assessed the efficacy of asparagine depletion using human cancer cell lines. In the present study, we evaluated the clinical relevance of asparagine depletion using a novel patient-derived spheroid xenograft (PDSX) mouse model. First, we examined ASNS expression in 38 spheroid lines and found that 12 lines (12/37, 32.4%) displayed high ASNS expression, whereas 26 lines (25/37, 67.6%) showed no ASNS expression. Next, to determine the role of asparagine metabolism in tumor growth, we established ASNS-knockdown spheroid lines using lentiviral short hairpin RNA constructs targeting ASNS. An in vitro cell proliferation assay demonstrated a significant decrease in cell proliferation upon asparagine depletion in the ASNS-knockdown spheroid lines, and this was not observed in the control spheroids lines. In addition, we examined asparagine inhibition with the anti-leukemia drug L-asparaginase (L-Asp) and observed a considerable reduction in cell proliferation at a low concentration (0.1 U/mL) in the ASNS-knockdown spheroid lines, whereas it exhibited limited inhibition of control spheroid lines at the same concentration. Finally, we used the PDSX model to assess the effects of asparagine depletion on tumor growth in vivo. The nude mice injected with ASNS-knockdown or control spheroid lines were administered with L-Asp once a day for 28 days. Surprisingly, in mice injected with ASNS-knockdown spheroids, the administration of L-Asp dramatically inhibited tumor engraftment. On the other hands, in mice injected with control spheroids, the administration of L-Asp had no effect on tumor growth inhibition at all. These results suggest that ASNS inhibition could be critical in targeting asparagine metabolism in cancers.

## 1. Introduction

RAS genes (*KRAS*, *HRAS*, and *NRAS*) are the most frequently mutated oncogenes in human cancers [1,2,3,4]. Colorectal cancer (CRC) is the fourth most common cause of cancer-related deaths, and about half of CRC patients have mutated RAS genes [1,2,3,4]. CRC cases harboring mutated RAS genes have been proven to be resistant to anti-epidermal growth factor receptor (EGFR) therapy in a number of clinical trials. Therefore, a new treatment strategy for RAS-mutated CRC is desired [5,6,7].

Metabolic changes in cancers were first observed by Warburg [8]. Recently, reprogramming of energy metabolism has been considered one of the hallmarks of cancer [9,10,11,12], and oncogenic RAS has been reported to be a key regulator in the reprogramming of metabolism [1,2,3,4,13]. Based on comprehensive gene expression analysis, *KRAS*-mutated CRC is classified into a molecular group characterized by metabolic dysregulation [14]. Recent advances in cancer metabolism research have suggested that metabolic changes caused by RAS mutations are potential therapeutic targets in *KRAS*-mutated CRC patients [4].

RAS mutation-induced metabolic reprogramming affects nutrient intake, glucose metabolism, and glutamine metabolism [4,13]. Oncogenic RAS can drive macropinocytosis in *KRAS*-mutated pancreatic ductal adenocarcinoma (PDCA) and CRC [15,16,17]. *KRAS*-mutated cancers scavenge extracellular constituents through macropinocytosis in order to meet their adaptation to nutrient stress. We recently reported that the dual blockade of asparagine bioavailability and macropinocytosis could represent a novel therapeutic strategy for *KRAS*-mutated cancers [17]. Regarding glucose metabolism, *KRAS*-mutated cancers increase glucose uptake by upregulating glucose transporters in CRC and PDCA [18,19,20,21]. Oncogenic KRAS controls glucose metabolism by altering key enzymes involved in anabolic glucose metabolism and increases glycolytic intermediates through pathways such as the hexosamine biosynthesis pathway and pentose phosphate pathway [19]. In addition to glucose metabolism, tumor cells are dependent on amino acids, especially glutamine [4,22,23,24]. When imported into the cells via the glutamine transporter, glutamine is converted to glutamate by glutaminase, then converted to the TCA cycle intermediate α-ketoglutarate by glutamate dehydrogenase 1 (GLUD1) or aminotransferases. Glutamine serves as a carbon source for the tricarboxylic acid (TCA) cycle and as a nitrogen source for nucleotides and amino acids; however, *KRAS*-mutated PDCA utilizes a novel pathway to produce NADH to maintain redox balance [23]. Glutamine is catalyzed to aspartate by glutaminase and mitochondrial aspartate aminotransferase 2 (GOT2), and aspartate is subsequently catalyzed to pyruvate by cytosolic aspartate aminotransferase 1 (GOT1), malate dehydrogenase 1, and malic enzyme 1. *KRAS*-mutated PDCA upregulates the expression of these enzymes, resulting in the production of NADPH in the cytoplasm, and this NADPH is used as a redox agent. On the other hand, we previously reported that *KRAS*-mutated CRC could facilitate another pathway of aspartate metabolism, wherein the cancer cells could become adaptive to glutamine depletion by upregulating the expression of asparagine synthetase (ASNS), an enzyme that synthesizes asparagine from aspartate, thereby avoiding nutrient stress-induced apoptosis [25]. Recently, asparagine metabolism has been extensively investigated as a new target for metabolic reprogramming in cancer [26,27,28]. Knott et al. reported that ASNS expression was correlated with metastatic relapse in breast cancer patients, and that asparagine bioavailability governed metastasis in a mouse model of breast cancer [26]. In KRAS-driven non-small-cell lung cancer (NSCLC), Gwinn et al. reported that mutated KRAS altered amino acid uptake and asparagine biosynthesis through ATF4 regulation under nutrient depletion, and that ASNS could contribute to apoptotic suppression, protein biosynthesis, and mTORC1 activation [27].

Our previous study assessed the efficacy of asparagine depletion by ASNS inhibition (for endogenous asparagine) and L-asparaginase (L-Asp), an anti-leukemia drug that converts asparagine to aspartate (for serum asparagine) [25]. Here, to further validate potential clinical relevance, we aimed to investigate the therapeutic effect of targeting ASNS using a patient-derived spheroid xenograft (PDSX) model, reported by Miyoshi et al. [29,30,31]. Patient-derived spheroid (PDS) can be propagated in vitro while retaining the genetic and morphological characteristics of the original tumors, and the PDSX model provides precise and predictable tumor growth for the assessment of chemosensitivity [31]. Therefore, we adapted the PDSX model to evaluate the effects of asparagine depletion, and investigated the effect of ASNS inhibition and L-Asp therapy on CRC in a pre-clinical setting.

## 2. Materials and Methods

### 2.1. Human Sample

Human colorectal samples were collected from patients who underwent surgery at Kyoto University Hospital. This study protocol was approved by the institutional review board of Kyoto University (R-0915), and patients provided their written consent for data handling.

### 2.2. Animal Experiments

Female KSN/slc nude mice (Japan SLC, Hamamatsu, Japan), age 8 to 11 weeks, were bred under specific pathogen-free conditions. The animal experiment protocol was approved by the Animal Care and Use Committee of Kyoto University (Med Kyo 20159).

### 2.3. PDS Culture

PDS cells were embedded in Matrigel (Corning, Corning, NY, USA) and cultured in the cancer medium, as previously reported [30,31]. We prepared an amino acid-reduced media based on components of the cancer medium. The medium was made from glutamine- and asparagine-free DMEM (Cat# 040-30095, FUJIFILM Wako Pure Chemical Corporation, Osaka, Japan) with insulin, transferrin, selenium, ethanolamine solution (ITS -X) (Thermo Fisher Scientific, Waltham, MA, USA), 10% fetal bovine serum, 1% penicillin/streptomycin, 10 mM Y-27632 (TOCRIS Bristol, Bristol, UK), and 10 mM SB-431542 (TOCRIS).

### 2.4. Preparation of Protein for Western Blotting

Spheroids in Matrigel were suspended in Cell Recovery Solution (Corning) and digested with rotation for 30–60 min. Afterwards, they were centrifuged and washed with cold PBS twice. Spheroids were lysed in SDS lysis buffer (70 mM Tris-HCL, 3% SDS, 10% Glycerol). As primary antibodies for ASNS, rabbit polyclonal anti-ASNS (Abcam, Cambridge, UK) was used.

### 2.5. Quantitative Reverse Transcription Polymerase Chain Reaction (RT-PCR) Analysis

Total RNA was extracted as previously reported with a high purity RNA isolation kit (Roche, Basel, Switzerland). Reverse transcription was performed with RevaTra Ace (TYOBO, Osaka, Japan) according to the manufacture’s protocol. The synthesized cDNA was quantified with StepOnePlus Real-Time PCR system (Thermo Fisher Scientific, Waltham, MA, USA) and THUNDERBIRD SYBR qPCR Mix (TOYOBO, Chuo City, Tokyo). The following primers were used: ASNS, 5′- CAGCTGAAAGAAGCCCAAGT-3′ and 5′- AGAGCCTGAATGCCTTCCTC-3′; b-actin, 5′-GCAAAGACCTGTACGCCAAC-3′ and 5′-ACATCTGCTGGAAGGTGGAC-3′.

### 2.6. Construction of Recombinant Lentivirus

To knock down ASNS expression in spheroids, we used the following oligonucleotides: shASNS#1-sense, 5′-CCGGATGGTGAAATCTACAAC CATACTCGAGTATGGTTGTAGATTTCACCATTTTTTG-3′; shASNS#1-antisense, 5′-AATTCAAAAAATGGTGA AATCTACAACCATACTCGAGTATGGTTGTAGATTTCACCAT-3′; shASNS#2-sense, 5′- CCGGTTAGGTGGTCTTTATGCTG TACTCGAGTACAGCATAAAGACCACCTAATTTTTG-3′; shASNS#2-antisense, 5′- AATTCAAAAATTAGGTGGTCTTTATG CTGTACTCGAGTACAGCATAAAGACCACCTAA-3′. To use EGFP as a selection marker, we constructed EGFP-introduced pLKO.1-TRC control vector (pLKO.1 EGFP scramble) that transduced the EGFP gene from pCX-EGFP to the puromycin resistance site of pLKO.1 puro scrample. Each set of oligonucleotides was annealed and cloned into the AgeI/EcoRI sites of pLKO.1 EGFP.

### 2.7. Transfection with Lentivirus to Spheroids

We produced the lentivirus to introduce pLKO.1 EGFP vector cloned with shASNS using lentiviral packaging plasmid psPAX2 and VSV-G envelope expressing plasmid pMD2.G. Generation of lentiviral particles was made by 293FT cells transfected using these plasmid vectors. Then, we produced a concentration of the lentiviral particles with PEG-it Virus Precipitation Solution. To infect a spheroid with lentivirus, trypsinized spheroids were incubated with the lentiviral concentration for 6 h, then these spheroids were embedded into Matrigel. After 3 to 4 days of culturing, we scratched and suspended the embedded spheroids and picked up GFP-positive spheroids under a fluorescence microscope to select the transfected spheroids. We repeated this selection step until all cells expressed GFP to obtain spheroids that were almost transfected.

### 2.8. Cell Proliferation Assay

After trypsinization, we created a spheroid suspension and passed it through 40 μm cell strainer to exclude irregular large cell aggregates. After centrifugation and re-suspension, we checked the density and viability of the suspension. For the cell proliferation assay, we prepared 96-well plates in which the spheroid (2000 aggregates/well) was embedded in Matrigel (4 μL/well) and cultured spheroids with the cancer media or an amino acid reduced media (100 μL/well). The cell proliferation assay was conducted using a Cell Counting Kit-8 (CCK-8) Dojin, Kumamoto, Japan) according to the manufacturer’s protocol. We analyzed the absorbance (460 nm) and calculated the relative absorbance rate at day 3 to day 0.

### 2.9. Immunohistochemistry

We removed the Matrigel from the cultured spheroids with a Cell Recovery Solution (Corning). A paraffin-embedded spheroid sample was produced using iPGell (GenoStaff: PG20-1, GENOSTAFF, Tokyo, Japan) from the Matrigel-removed spheroids according to the manufacturer’s protocol. Formalin-fixed and paraffin-embedded sections of spheroids and CRC tissues from which spheroids were established were stained with anti-rabbit ASNS (Abcam, ab40850), anti-mouse Ki67 (Dako Agilent, Santa Clara, CA, USA), or anti-Cleaved Caspase-3 (Cell Signaling, Danvers, MA, USA). Antigen retrieval was achieved with a microwave in citrate buffer (pH: 6.0). For ASNS expression in clinical samples, the ASNS immunoreactivity score was determined by the sum of distribution and staining intensity, as previously described [16]. For ASNS expression in spheroids, we used only the staining intensity and defined no staining as low expression and weak to strong staining as high expression, because it was difficult to assess the distribution in very small areas of spheroids.

### 2.10. Statistical Analysis

Analyzed values are expressed as mean values ± standard error of the mean (SEM). Continuous variables were determined by Student’s *t*-test, as appropriate. All in vitro experiments were performed at least three times. All analyses were two-sided, and a *p* value <0.05 was considered to be statistically significant. Statistical analyses were performed using JMP Pro software (SAS Institute Inc., Cary, NC, USA).

## 3. Results

### 3.1. Patient-Derived Spheroid (PDS) Partially Express ASNS In Vitro

To assess the expression of ASNS in CRC PDS, we examined the expression level of ASNS in 37 established spheroid lines using immunohistochemistry. We observed that 32.4% (12/37) of the in vitro spheroid cultures displayed a high expression of ASNS, whereas 67.6% (25/37) displayed a low expression (Figure 1A). In addition, we investigated the relationship between *KRAS* status and ASNS expression, finding that ASNS expression was high in 36.4% (8/22) of spheroids with mutated *KRAS* and in 26.7% (4/15) of spheroids with wild-type *KRAS*, which indicates that there was no significant correlation between *KRAS* status and ASNS expression in the spheroid cultures. We previously reported a high ASNS expression in 50% (46/93) of primary CRC tissues [25]. Owing to the discrepancy in the positive rate of ASNS expression between CRC tissues and spheroids, we evaluated ASNS expression in the primary CRC tissues from which each of these spheroids was established (Figure 1B). ASNS expression was high in 51.4% (19/37) of the CRC tissues (Figure 1C), indicating an increased rate of high ASNS expression in CRC tissues compared to that in spheroids. ASNS expression was high in 50% (11/22) of CRC tissues with mutated *KRAS* and in 33% (5/15) of CRC tissues with wild-type *KRAS*. Considering that ASNS expression is possibly controlled by nutrient status [25], this discrepancy might be caused by differences in nutrient status between in vitro culture conditions and the in vivo tumor microenvironment.

To evaluate the role of asparagine metabolism in tumor growth using PDS, we inhibited de novo asparagine synthesis using lentiviral short hairpin RNA (shRNA) constructs targeting *ASNS* (referred to as sh*ASNS* #1 and sh*ASNS* #2). Two spheroid lines, HC27T and HC17T, were selected based on ASNS expression (Figure 2A). Green fluorescent protein (GFP) was used as a selection marker to confirm the transduction of shRNA construct into the spheroids. Effective shRNA transduction into spheroids was confirmed by producing small cell aggregates of spheroids with trypsinization and selecting some with only GFP-positive cells (Figure 2B). After repeating the procedure two or three times, we established ASNS-knockdown spheroid clones of these two spheroid lines (Figure 2C,D).

### 3.2. Asparagine Depletion Inhibits Spheroid Growth In Vitro

Next, we analyzed cellular responses to asparagine depletion by inhibiting the exogenous and endogenous asparagine supply. We conducted cell proliferation assays under asparagine depletion with or without inhibition of de novo asparagine synthesis by ASNS-knockdown. In the control spheroids lines, asparagine depletion did not affect cell proliferation. However, asparagine depletion significantly decreased cell proliferation in ASNS-knockdown spheroids (Figure 3A). In addition, we examined the effect of asparagine depletion by L-Asp, an FDA-approved key drug for acute lymphoblastic leukemia (ALL). In the ASNS-knockdown spheroid lines, substantial inhibition of cell growth was observed even with a low concentration of L-Asp (0.1 U/mL), whereas the same concentration displayed limited growth inhibition in control spheroid lines (Figure 3B). However, at high concentrations (>1 U/mL) L-Asp inhibited the growth of control spheroid lines by up to 20%. These results indicate that asparagine is a key metabolite in CRC cell proliferation and that ASNS-knockdown could enhance the anti-tumor effects of L-Asp.

### 3.3. ASNS Inhibition Is Critical for Targeting Asparagine Metabolism In Vivo

To assess the effect of asparagine depletion on tumor growth in vivo, we used a PDSX model. After inoculation of one spheroid line (HC17T) into nude mice, we continuously administered L-Asp to deplete serum asparagine from day 1 post-inoculation (Figure 4A–C). Surprisingly, L-Asp treatment completely suppressed tumor growth in ASNS-knockdown xenografts until day 28 post-inoculation. Growth inhibition continued even after L-Asp treatment was terminated. In contrast, L-Asp treatment did not inhibit tumor growth in control xenografts. Rather, tumor growth was enhanced by L-Asp injection in control xenografts. Furthermore, we confirmed that ASNS expression was sufficiently knocked down in the xenografts (data not shown).

To further investigate the effectiveness of the combination of ASNS inhibition and L-Asp, we injected L-Asp after the ASNS-knockdown xenografts reached an average volume of 70 mm^3^ (Figure 5). Compared with vehicle-treated control xenografts, those from ASNS-knockdown xenografts treated with L-Asp exhibited significantly suppressed growth rates. These results suggest that ASNS inhibition is the most important factor for asparagine-targeted therapy.

## 4. Discussion

Nutrient limitation has been recognized as a potential therapeutic strategy for cancer. Many cancer cells are metabolically reprogrammed to shift from catabolic to anabolic glutamine utilization. Therefore, the mechanisms by which cancer cells adapt to glutamine depletion have been investigated. Asparagine is not catabolized by mammalian cells, and is known to play a pivotal role in tumor progression. In certain cell types, asparagine is necessary for adaptation to glutamine depletion-induced apoptosis [32,33]. Furthermore, asparagine is involved in the exchange of extracellular amino acids such as arginine, serine, and histidine [34]. ASNS is an enzyme that synthesizes asparagine from aspartate using glutamine. Here, we describe the potency of asparagine metabolism-targeted therapy using PDS or PDSX in a preclinical setting. When ASNS-knockdown PDS were treated with different concentrations of L-Asp in vitro, synergistic anti-proliferative effects of L-Asp were observed even at a low concentration (0.1 U/mL) (Figure 3B). In addition, the combination of ASNS inhibition and L-Asp treatment displayed a synergistic anti-proliferative effect in the PDSX model (Figure 4 and Figure 5). In this study, we used shRNA-based ASNS interference as a method of ASNS inhibition, as there is no available chemical that inhibits the enzyme efficiently. To our knowledge, there is only one inhibitor of ASNS, a slufoximine-based inhibitor which is a stable analog of the transition state in asparagine synthesis, which inhibits ASNS in a competitive manner [35]. The poor cell permeability of this compound makes it impossible to evaluate the efficacy of ASNS inhibition in vivo. Recently, another target for inhibiting ASNS was reported. Nakamura et al. showed that general control nonderepressible 2 (GCN2) governed the cellular response to amino acid limitation and that GCN2 inhibition sensitized ASNS-low cancer cells to L-Asp both in vitro and in vivo [36]. They produced a novel GCN2 inhibitor and showed its inhibitory effect in L-Asp-resistant ALL cell lines with elevated ASNS expression. This line may be a candidate for therapeutic targeting of asparagine metabolism.

In this study, we did not examine how the other metabolites (i.e., aspartate and α-ketoglutarate) could affect tumor growth with other pathways. The established ASNS-knockdown spheroid lines should be considered as specific cells which obtained tolerance to ASNS suppression using a specific pathway to compensate for the metabolic demand. Considering the aspect of asparagine metabolism, aspartate is an important metabolite. In PDCA, KRAS mutations regulate the glutamine metabolism through its conversion to aspartate, thereby supporting growth by maintaining the cellular redox balance [23]. While most cells utilize GLUD1 to convert glutamine-derived glutamate into α-ketoglutarate, PDCA cells metabolize glutamine through an unconventional pathway in which glutamine is converted to non-essential amino acids such as aspartate by GOT1. On the other hand, we previously reported that mutated KRAS did not alter the expression of GLUD1 or GOT1 in CRC [25], indicating that the multifaceted roles of mutated KRAS in metabolism are cell type-dependent. Aspartate plays a role in mitochondrial respiration, and exogenous aspartate addition could restore proliferation of respiration-deficient cancer cells [37,38]. When treated with L-Asp, which catalyzes asparagine to aspartate, circulating aspartate can be sustained at the increased concentration. In the present study, accelerated tumor growth during L-Asp treatment without ASNS inhibition was observed (Figure 4A). This tumor growth might be explained by the increased level of circulating aspartate. Further investigation is needed to evaluate the strategy targeting ASNS in CRC treatment using other methods, such as a conditional knockdown/knockout system in cell-based assays.

The correlation between ASNS expression and KRAS mutations could not be elucidated. While PDSX has been established from the CRC tissue samples under specific culture conditions, these differ from the metabolic environment of CRC tumors in patients. This difference could mask the characteristics of tumor metabolism in vivo. More sophisticated in vivo culture methods to represent the metabolic environment are needed in order to assess the metabolic changes in this PDSX system.

In light of the reliance of cancer cells on metabolic changes for their growth and survival, targeting metabolic processes could be a potential therapeutic strategy. A new strategy for the treatment for *KRAS*-mutated CRC is needed to improve the prognosis of advanced CRC. *KRAS* mutation coordinates the multifaceted reprogramming of cancer metabolism; therefore, the network of this reprogramming should be elucidated in order to develop novel drugs for *KRAS*-mutated cancers.

## Figures and Tables

**Figure 1 cells-11-03273-f001:**
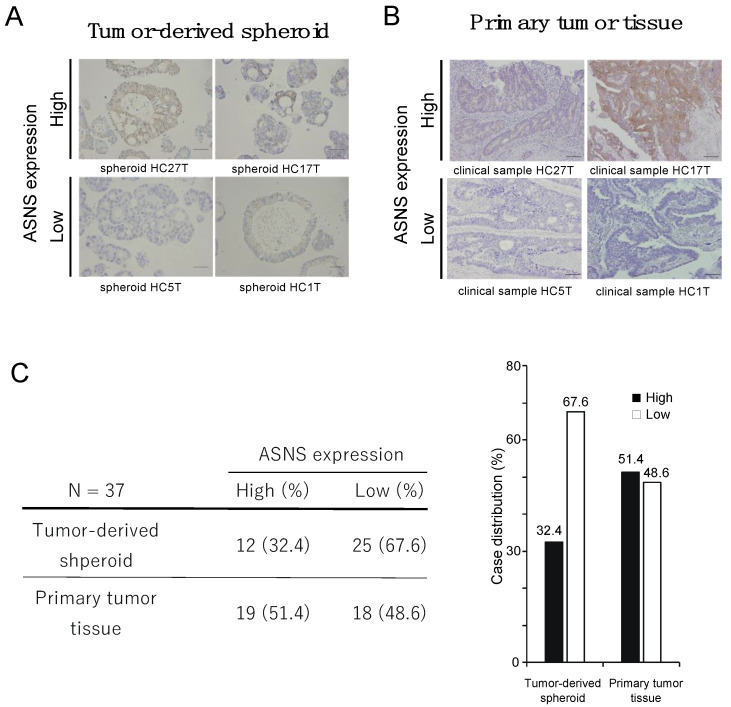
ASNS expression in CRC. (**A**) H&E staining for ASNS for tumor-derived spheroids. Scale bars, 100 µm. (*B*) H&E staining for ASNS for primary CRC tissues. Scale bars, 100 µm. (**C**) Status of ASNS expression.

**Figure 2 cells-11-03273-f002:**
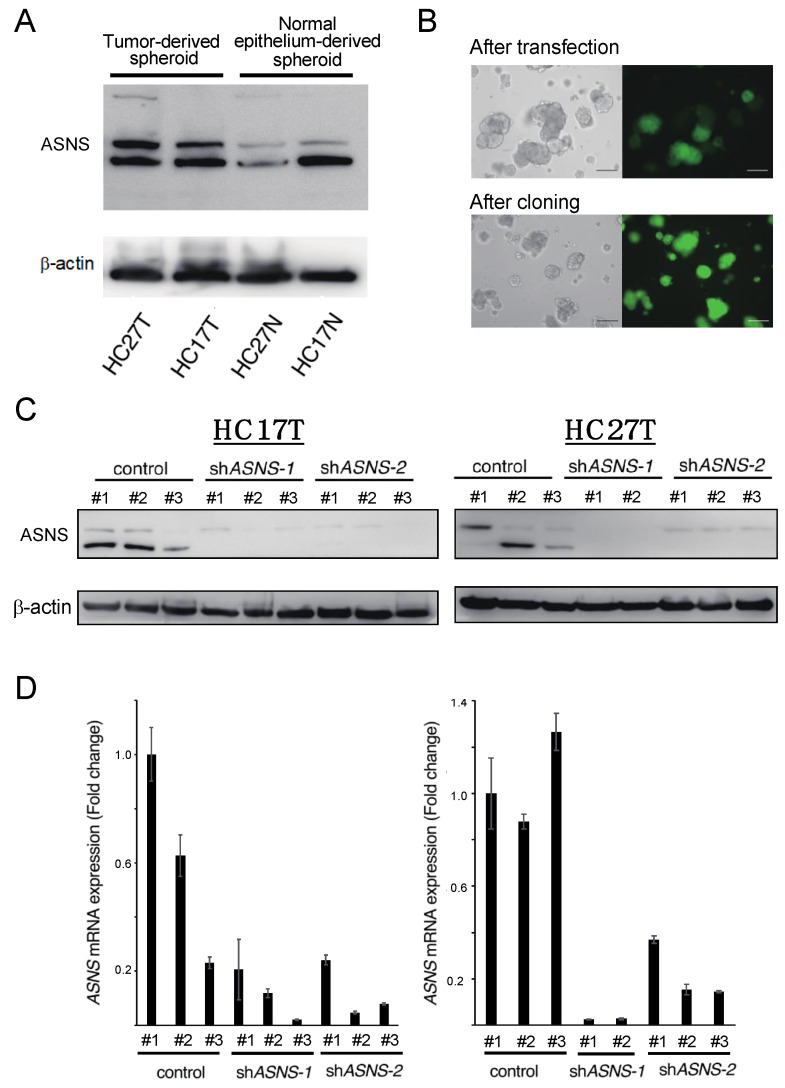
Construction of ASNS-knockdown spheroid clones. (**A**) Western blot analysis of ASNS expression. (**B**) GFP was used as a selection marker to confirm the transduction of shRNA construct into the spheroids. Scale bars, 100 µm. (**C**) Western blot analysis of ASNS expression. Two spheroid lines, HC27T and HC17T, were treated with independent shRNA constructs targeting ASNS. (**D**) Quantitative RT-PCR showing relative mRNA levels for ASNS. The # in figure means each estab-lished clone.

**Figure 3 cells-11-03273-f003:**
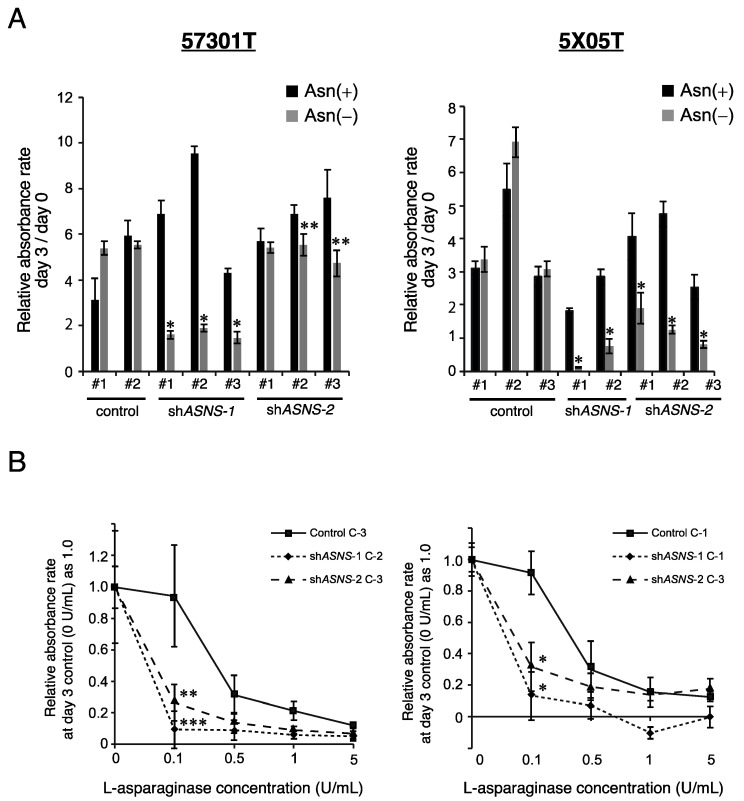
Inhibition of ASNS suppresses spheroid growth in vitro. (**A**) Cell proliferation measured by CCK-8 assay. The bar graph shows the relative absorbance rate on day 3 compared with day 0. Student’s *t* test, *p* * < 0.01, *p* ** < 0.05. (**B**) Cell proliferation measured by CCK-8 assay. Cells transfected with control or two independent sh*ASNS* vectors were treated with various concentrations of L-Asp. Student’s *t* test, *p* * < 0.01, *p* ** < 0.05, *p* *** = 0.05.

**Figure 4 cells-11-03273-f004:**
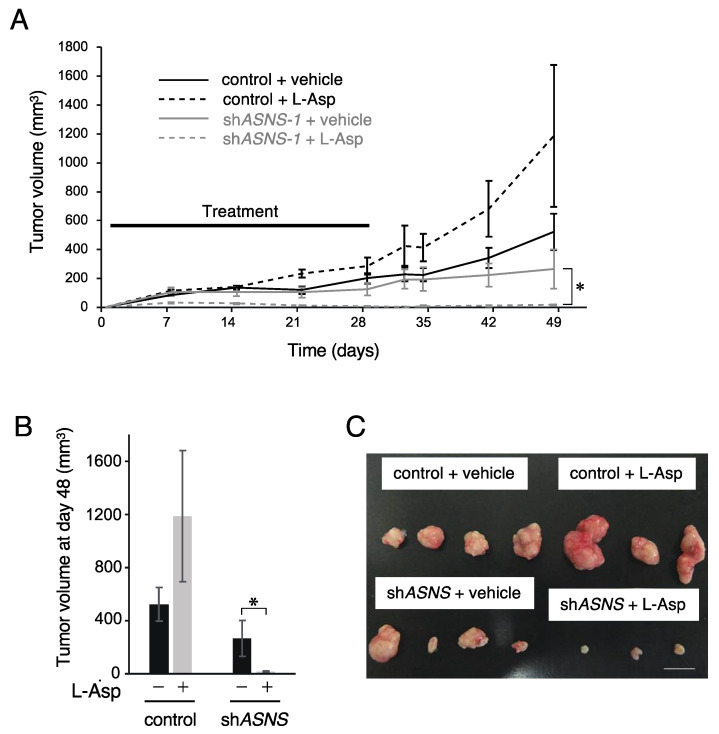
Inhibition of ASNS suppresses spheroid growth in vivo. (**A**) Xenograft size of control or ASNS-knockdown spheroids with vehicle or L-Asp. (**B**) Xenograft size at day 48 post-inoculation. Mean; bar, ±SEM, *n* = 5–7 mice in each group. * *p* < 0.05 by Student’s *t*-test. (**C**) Representative images of tumors are shown. Scale bars, 1 cm.

**Figure 5 cells-11-03273-f005:**
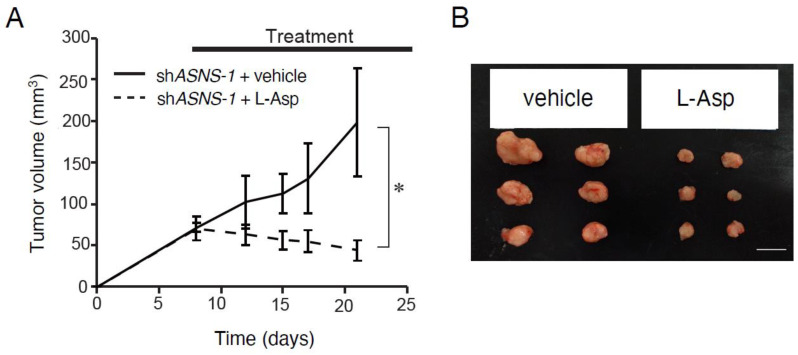
Inhibition of ASNS suppresses spheroid growth in vivo. (**A**) Xenograft size of ASNS-knockdown spheroids with vehicle or L-Asp. Mean; bar, ±SEM, *n* = 5–7 mice in each group. * *p* < 0.05 by Student’s *t*-test. (**B**) Representative images of tumors. Scale bars, 1 cm.

## Data Availability

The data presented in this study are available from the corresponding author upon reasonable request.

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
