# Peer review of "Targeting Asparagine Synthetase in Tumorgenicity Using Patient-Derived Tumor-Initiating Cells"

_cells, 2022, doi:10.3390/cells11203273_

Round 1
Reviewer 1 Report
Please see the comments in the PDF.

Author Response
To Reviewer 1:
As requested, we have added more 10 references in the revised manuscript. Furthermore, we have modified the Discussion, as requested by Reviewer 1.
Other comments were revised as necessary. However, the revision period is only 10 days, so we hope the reviewer 2 will take that into consideration.

Reviewer 2 Report
This manuscript demonstrated that the role of shASNS in cancer cells in vivo and in vitro.The manuscript provided some evidence that ASNS could serve as a potential target for CRC treatment at some distend. However, digging more mechanism will make this manuscript stronger.
Specific comments:
1. In figure 2C, there are 2 bands for ASNS, can the authors specify which is the major band or both 2 bands are all ASNS?
2. How about the cellular asparagine level after knock down the ASNS in cells?
3. Does asparagine depleted medium have the same effect to cells?
4. Does adding extra asparagine rescue the shASNS effect?
5. How about the ASNS level in xenograft tumors?
6. Does knocking down ASNS affect other metabolites? Like glutamate and TCA cycle?
7. Does knock down ASNS affect metastasis?
Author Response
To Reviewer 2:
Reviewers’ comments are quoted “verbatim” >> followed by our response.
“ COMMENTS FOR THE AUTHOR:
Reviewer 2: This manuscript demonstrated that the role of shASNS in cancer cells in vivo and in vitro. The manuscript provided some evidence that ASNS could serve as a potential target for CRC treatment at some distend. However, digging more mechanism will make this manuscript stronger.”
>> We thank Reviewer 2 for the positive comment.
“ Specific comments:
- In figure 2C, there are 2 bands for ASNS, can the authors specify which is the major band or both 2 bands are all ASNS? ”
>> Thank you for your comment. We consider that both 2 bands are ASNS. This is based on the fact that ASNS-knockdown experiments confirmed both bands disappeared in clones with high knockdown efficiency at the mRNA level (Fig. 2C and 2D). As to why 2 bands were produced, we believe it was because the processing of the protein samples caused some degradation.
“ 2. How about the cellular asparagine level after knock down the ASNS in cells? ”
>> Cellular asparagine concentrations were not measured in this study. Fig 3A showed that, under asparagine depletion, there was a significant difference in cell proliferation between control spheroids and ASNS-knockdown spheroids. This effect was not observed under asparagine-containing medium. Therefore, we assume that intracellular asparagine is compensated by uptake of asparagine, and that there is no significant difference in intracellular asparagine level between control spheroids and ASNS-knockdown spheroids
“ 3. Does asparagine depleted medium have the same effect to cells? ”
>> Yes. In Fig 3A, we observed that asparagine depleted medium significantly decreased cell proliferation in ASNS-knockdown spheroids. In Fig 3B, we also observed the inhibitory effect of asparagine depletion by L-Asp.
“ 4. Does adding extra asparagine rescue the shASNS effect? ”
>> Yes. In Fig 3A, the inhibitory effect of cell proliferation in ASNS-knockdown spheroids (shASNS-1 and shASNS-2) was not observed under asparagine-containing medium.
“ 5. How about the ASNS level in xenograft tumors? ”
>> We appreciate the comment. We confirmed that ASNS expression was sufficiently knocked down in xenografts by immunostaining. We have added the data as Fig-A at the end of this rebuttal. We have also added this point in the revised text (page 8, lines 258-259).
“ 6. Does knocking down ASNS affect other metabolites? Like glutamate and TCA cycle? ”
>> The effects on the TCA cycle and other metabolites were not verified in this study. It will be interesting to examine this point in the future.
“ 7. Does knock down ASNS affect metastasis? ”
>> The effects of ASNS-knockdown on metastasis were not verified in this study. I agree that this point is worth investigating in the future.
We corrected the text under the help of English-native consultants (Editage; www.editage.jp). At the end of this rebuttal, we have added a certificate of English editing that was issued from Editage (www.editage.jp).

Round 2
Reviewer 2 Report
The questions are properly responded. I do not have further questions and recommend acceptance.
Author Response
Comments to the original manuscript (cells-1926400)
Reviewer 2’s comments are quoted “verbatim” >> followed by our response.
“The questions are properly responded. I do not have further questions and recommend acceptance.”
>> Thank you for the positive comment.
